# Usefulness of Perioperative Nutritional Therapy with the Glutamine/Arginine/Calcium β-Hydroxy-β-Methylbutyrate Product in Esophageal Cancer Surgery: A Single-Center Retrospective Study

**DOI:** 10.3390/nu16132126

**Published:** 2024-07-03

**Authors:** Koichi Okamoto, Hiroyuki Takamura, Taigo Nagayama, Yuta Sannomiya, Akifumi Hashimoto, Hisashi Nishiki, Daisuke Kaida, Takashi Miyata, Toshikatsu Tsuji, Hideto Fujita, Shinichi Kinami, Itasu Ninomiya, Noriyuki Inaki

**Affiliations:** 1Department of General and Digestive Surgery, Kanazawa Medical University Hospital, 1-1 Daigaku, Uchinadamachi, Kahoku 920-0293, Ishikawa, Japan; takamuh@kanazawa-med.ac.jp (H.T.); nagayama@kanazawa-med.ac.jp (T.N.); stttty@kanazawa-med.ac.jp (Y.S.); ah-idai@kanazawa-med.ac.jp (A.H.); hisashi@kanazawa-med.ac.jp (H.N.); kaida-d@kanazawa-med.ac.jp (D.K.); t-miyata@kanazawa-med.ac.jp (T.M.); hfujita@kanazawa-med.ac.jp (H.F.); 2Department of Gastrointestinal Surgery, Kanazawa University, 13-1 Takara-Machi, Kanazawa 920-8641, Ishikawa, Japan; tsuji104@staff.kanazawa-u.ac.jp (T.T.); n.inaki@med.kanazawa-u.ac.jp (N.I.); 3Department of General and Digestive Surgery, Kanazawa Medical University Himi Municipal Hospital, Himi, Toyama 935-8531, Japan; kinami@kanazawa-med.ac.jp; 4Department of Surgery, Fukui Prefectural Hospital, 2-8-1 Yotsui, Fukui 910-0846, Japan; nino@staff.kanazawa-u.ac.jp

**Keywords:** esophageal cancer, minimally invasive esophagectomy, complication, sarcopenia

## Abstract

A useful perioperative nutritional therapy for highly invasive esophageal cancer surgical cases needs to be developed. We clarified the usefulness of amino-acid-enriched nutritional therapy using glutamine (Gln)/arginine (Arg)/calcium β-hydroxy-β-methylbutyrate (HMB) products on the short-term postoperative outcomes of minimally invasive esophagectomy for esophageal cancer. Altogether, 114 patients (Gln/Arg/HMB group) received perioperative nutritional therapy with Gln/Arg/HMB products, and we retrospectively investigated the change in nutritional parameters including skeletal muscle mass, occurrence of postoperative complications, and short-term postoperative outcomes in this group. The results were compared between the Gln/Arg/HMB and control groups (79 patients not receiving the Gln/Arg/HMB products). The incidence of all postoperative complications, sputum expectoration disorder, and pleural effusion of grade ≥ III was significantly lower in the Gln/Arg/HMB group (62.0% vs. 38.6%, *p* = 0.001; 44.3% vs. 28.1%, *p* = 0.020; 27.8% vs. 13.2%, *p* = 0.011, respectively). The psoas muscle area and postoperative body weight were significantly higher at 1 month and 1 year after surgery in the Gln/Arg/HMB group than in the control group (93.5% vs. 99.9%, *p* < 0.001; 92.0% vs. 95.4%, *p* = 0.006). Perioperative amino-acid-enriched nutritional therapy may improve the short-term postoperative outcomes, nutritional status, and skeletal muscle mass of esophageal cancer surgical patients.

## 1. Introduction

Esophageal cancer surgery is a highly difficult and invasive procedure that involves esophagectomy with mediastinal lymph node dissection, abdominal manipulation with gastric tube creation, and anastomosis by cervical manipulation [1]. In Japan, the surgical mortality rate of esophageal cancer surgery is 3.2%, and the complication rate is also reported to be very high at 41.9%, despite the recent popularity of minimally invasive esophagectomy (MIE). Advanced age, poor physical status (PS), chronic obstructive pulmonary disease, smoking, and malnutrition are the reported risk factors for death from surgery [1]. Additionally, the presence of sarcopenia and nutritional disorders has a considerable impact on the outcomes, including complication rates and long-term prognosis, in the treatment of various types of cancer [2,3]. Given that most esophageal cancer patients develop sarcopenia, effective interventions for nutritional management and rehabilitation are essential to prevent complications during the perioperative period [4]. Contrarily, the concept of “pharmaconutrition”, which administers nutrients according to the pathology, is important, and several reports have reported on the usefulness of various pharmaconutrients in nutritional interventions following surgery for gastrointestinal cancers [5,6]. However, only a few reports have described the effectiveness of amino-acid-rich perioperative nutritional therapy on the management of patients who had undergone esophageal cancer surgery.

Thus, the present study aimed to investigate whether the postoperative outcomes, including complication rate, nutritional status, and skeletal muscle mass, can be improved with amino-acid-rich perioperative nutritional therapy using glutamine (Gln)/arginine (Arg)/calcium β-hydroxy-β-methylbutyrate (HMB) products as pharmaconutrition in the perioperative management of patients who had undergone esophageal cancer surgery.

## 2. Materials and Methods

### 2.1. Patients

We enrolled 193 patients who underwent MIE at Kanazawa University Hospital between September 2004 and June 2019. The exclusion criteria of this study were patients who underwent two-stage esophagectomy, those who had preoperative radiotherapy, or those with missing data for evaluation. All patients were staged according to the Union for International Cancer Control TNM staging, version 8 [7]. The clinico-oncological characteristics of the patients in both groups are compared with respect to age, sex, performance status (PS), American Society of Anesthesiologists PS (ASA-PS), tumor stage of esophageal cancer (cT, cN, cM, cStage), presence or absence of neoadjuvant chemotherapy (NAC), abdominal surgical approach (open laparotomy vs. hand-assisted laparoscopic surgery), fields of lymph node dissection (D2 vs. D3), reconstruction route (posterior mediastinal vs. retrosternal route), reconstructed organ (gastric conduit vs. small intestine), physical characteristics [height, weight, body mass index (BMI)], and blood tests results [white blood cell count (WBC), lymphocyte, C-reactive protein (CRP), albumin (Alb), and total cholesterol]. All data were collected and analyzed retrospectively. Our institutional ethics committee approved the research (Registry Number 1836).

### 2.2. Surgical Procedure for Thoracoscopic Esophagectomy

All the patients underwent MIE and reconstruction, as described elsewhere [8]. During the thoracic procedure in the left decubitus position, the patients were intubated with a double- or single-lumen endotracheal tube with a balloon blocker for one-lung ventilation. In 60 cases, artificial pneumothorax with CO_2_ insufflation at a pressure of 8–12 mmHg was used. Thoracoscopic esophagectomy with mediastinal lymph node dissection was performed via a 5 cm mini-thoracotomy and four 12 mm trocars or six ports without mini-thoracotomy. The dissection of the lymph nodes around the bilateral recurrent laryngeal nerves, trachea, and aorta was performed with caution. Abdominal and supraclavicular cervical lymph node dissection were performed simultaneously in the supine position. The digestive reconstruction with a gastric conduit via the mediastinal route was selected as the primary reconstruction method. In the cervical procedure, cervical anastomosis with or without cervical lymph node dissection was performed according to the tumor stage, location, and patients’ condition. In cases of noncurative resection, the retrosternal route reconstruction was selected to allow for subsequent chemoradiotherapy for the residual tumor. The small intestinal conduit was used in patients who had previously undergone gastrectomy or who required total gastrectomy for the treatment of simultaneous gastric cancer. All patients underwent enterostomy via a gastric tube during surgery [9]. The Clavien–Dindo classification was used to categorize the surgical morbidities [10].

### 2.3. Overview of Postoperative Tube Feeding and Physical Management

All patients were given early tube feeding with a basal enteral nutritional supplement from the day after surgery. Additionally, since May 2012, as perioperative pharmaconutrition for esophageal cancer surgery, patients were basically given two packets of Gln/Arg/HMB product (Abound^®^ containing Gln 7 g, Arg 7 g, and HMB 1.5 g, Abbott Nutrition, Columbus, OH, USA) per day starting at least 14 days preoperatively. Postoperatively, the patients were given the same amount of Gln/Arg/HMB product dissolved in warm water and administered via a jejunostomy tube along with basal enteral nutritional supplements from the day after surgery (Gln/Arg/HMB group). In contrast, patients who did not receive any Gln/Arg/HMB product (historical controls) were defined as the control group. As the basal enteral nutritional agent, MEIN^®^ (Meiji, Tokyo, Japan), which contains fat, was administered from September 2004 to March 2013, and Elental^®^ (EA Pharma, Tokyo, Japan), which does not contain fat, was routinely administered from April 2013 onward. Owing to the differences in formulations, the amount of fat compound between the basal nutrients differed, but the calories and fluids administered were almost the same. The flow rate of enteral nutrition was increased daily, and a nutritional plan was made to administer the nutritional agent equivalent to the patient’s energy requirements by the 7th day after surgery. Enhanced recovery after surgery (ERAS) was introduced as perioperative management other than enteral nutrition, and perioperative rehabilitation, oral care, synbiotics, etc., were routinely performed in all cases. The Gln/Arg/HMB product group comprised 114 patients, and the control group comprised 79 patients who had not received any Gln/Arg/HMB product. The inclusion criteria were as follows: (I) patients aged 18 years or over; (II) those who received preoperative blood sampling and computed tomography (CT) scan; (III) patients who underwent MIE. The exclusion criteria were as follows: (I) patients without enough data; (II) patients who died during perioperative period. In both groups, the patients’ background, surgical factors, type and severity of postoperative complications, length of hospital stay after surgery, nutritional parameters, body weight, and changes in the psoas muscle area were measured by CT.

### 2.4. Measurement of Psoas Muscle Area and Definition of Iliopsoas Muscle Index

As one of the nutritional and sarcopenia indicators, we measured the cross-sectional area of the bilateral psoas muscles at the level of the third lumbar vertebrae using the manual trace method from the CT images taken before and after surgery. The sum was defined as “psoas muscle area (PMA)”. Additionally, the “psoas muscle index (PMI)”, which is the area of the iliopsoas muscle divided by the square of the height (m^2^), was calculated, and the evaluation of sarcopenia was defined using the sarcopenia standards criteria [11]. Based on the sarcopenia standards criteria, the presence or absence of sarcopenia was determined according to the sex-specific cutoff values of 6.36 and 3.92 cm^2^/m^2^ for men and women, respectively.

### 2.5. Statistical Analysis

Numerical results are expressed as the mean ± standard deviation. The χ^2^, Fisher’s exact, and Student’s *t*-tests were performed as appropriate to analyze the clinicopathological variables that we feel may cause treatment-related malnutrition in clinical practice and the incidences of postoperative complications. Statistical significance was assumed for *p* < 0.05. Some clinicopathological factors were selected for the univariate and multivariate analyses of the risk factors for the incidence of postoperative complications. Factors with *p* < 0.05 were defined as independent risk factors for morbidity after MIE. All analyses were performed using SPSS (IBM SPSS Statistics, version 25; IBM Corp., Armonk, NY, USA).

## 3. Results

### 3.1. Correlations between the Clinico-Oncological Features and Intraoperative Factors in the Gln/Arg/HMB Product and Control Groups

The clinico-oncological characteristics of the patients in both groups are shown in Table 1. As shown in Table 1, the comparison of each preoperative factor between the two groups showed no differences in age, sex, PS, ASA-PS, cT, cN, cM, cStage, presence or absence of NAC, abdominal surgical approach, fields of lymph node dissection, reconstruction route, reconstructed organ, height, weight, BMI, WBC, lymphocyte, CRP, Alb, total cholesterol, and so on. In the comparison between the two groups the amount of administered amino acids and fat differed as the basal enteral nutritional agent was administered either with or without Abound^®^ [amino acids, 5.0 vs. 4.7 g/100 mL (Elental^®^) added with 14 g/1 pack or 28 g/2 packs (Abound^®^); fat, 2.8 g/100 mL (MEIN^®^) vs. 0.17 g/100 mL (Elental^®^), respectively]. However, an almost-similar amount of energy [1.0 vs. 0.83 kcal/mL (Elental^®^) added with 79 kcal/1 pack or 158 g/2 packs (Abound^®^)] and water were administered in both groups.

The intraoperative factors of both groups are shown in Table 2. The two groups significantly differ in the amount of total blood loss, usage of robot-assisted surgery, and abdominal approach.

### 3.2. Effect of Perioperative Gln/Arg/HMB Products on the Incidence of Postoperative Complications after MIE

Table 3 shows the incidence of postoperative complications after MIE in both groups. The Gln/Arg/HMB group showed significant higher incidences of all complications, sputum expectoration disorder, and pleural effusion of grade ≥ III based on the Clavien–Dindo classification (the Gln/Arg/HMB group vs. the control group; 38.6% vs. 62.0%, *p* = 0.001; 28.1% vs. 44.3%, *p* = 0.020; 13.2% vs. 27.8%, *p* = 0.011, respectively). There was no significant difference in the duration of postoperative hospital stay in both groups (49.5 vs. 46.3 days, *p* = 0.659).

### 3.3. Univariate and Multivariate Analyses of the Clinicopathological Factors Associated with Overall Survival

Univariate analyses showed that fT3 or fT4, fN1 or higher, stage III or higher, R1 or more residual tumor, and higher preoperative CRP level were significantly correlated with a poor prognosis. However, in the multivariate analyses, only fN1 or higher and R1 or more residual tumor were found as the significant predictors of poor prognosis (Table 4). Additionally, no significant prognostic impact was observed in the occurrence of postoperative complications, presence of preoperative sarcopenia, and usage of Gln/Arg/HMB product (Table 5).

## 4. Discussion

Our findings showed the effectiveness of the amino-acid-enhanced perioperative nutritional therapy using Gln/Arg/HMB products on the short-term outcome and skeletal muscle mass after MIE.

Esophageal cancer surgery is a highly invasive procedure with a high incidence of postoperative complications, which can likely cause increased protein catabolism, inhibition of protein synthesis, nutritional deficiency due to gastrointestinal dysfunction, and skeletal muscle atrophy [2]. MIE using a thoracoscopy can be expected to improve the short-term outcomes owing to its low invasiveness and reduced risk for complications as compared to the previous open esophagectomy approach [8,12,13,14]. Various efforts to prevent complications during the perioperative period of patients receiving esophageal cancer surgery have been reported. The use of intraoperative nerve monitoring and ingenuity in mediastinal lymph node dissection are useful to prevent recurrent laryngeal nerve paralysis, and these approaches also resulted in a reduction in the incidence of aspiration pneumonia, respiratory failure, and sputum expectoration disorders [15,16]. Additionally, the development of preoperative risk assessment based on the patient’s nutritional status and background diseases, intraoperative evaluation of the blood flow of the reconstructed organs using indocyanine green, and strict blood sugar management in diabetic patients have been reported to be useful in preventing the development anastomotic leakage [1,17,18,19,20]. The occurrence of various complications reportedly can lead to an increase in postoperative hospital stay, deterioration in nutritional status, and poor long-term outcomes [21,22]. From the viewpoint that the occurrence of disuse syndrome and worsening sarcopenia after surgery considerably impedes the return to daily life and delays the transition to additional cancer therapy, it is strongly desirable to prevent the development of all complications by taking preventive measures [23,24]. Contrarily, various studies reported on the positive efforts regarding effective measures to prevent the worsening of secondary sarcopenia after digestive cancer surgery [25,26,27]. Sarcopenia not only affects the quality of life (QOL) and activities of daily living (ADL) of patients with malignant tumors but also influences the occurrence of various treatment-related adverse events and complications; therefore, sarcopenia should not be ignored to ensure the safety and continuity of cancer treatment [22].

Patients with esophageal cancer have a high prevalence of malnutrition because of the presence of dysphagia caused by the tumor, along with secondary anorexia and cancer cachexia. Significant weight loss and nutritional deficiencies predispose a patient to an increased risk of complications and protracted admission [23]. Sarcopenia and frailty from cancer cachexia also affect patient outcomes and recovery [21,28]. Hence, it is essential to assess the nutritional status of the patient at diagnosis. The decision to intervene is based on the risk assessment, but nutritional supplementation is known to have a positive effect on the perioperative outcomes [29,30].

The mechanisms of progression of postoperative sarcopenia can be broadly summarized into three. The first is the delay or decrease in protein synthesis during postoperative wound healing. Nutrients, including amino acids and carbohydrates, administered to repair damaged tissues are mostly mobilized for tissue repair, and the distribution of nutrients to the skeletal muscles throughout the whole body is relatively reduced, possibly impairing the maintenance of skeletal muscle mass and its quality [31]. To correct this, the efficient administration of appropriate amino acids to promote protein synthesis is required. The calcium HMB, L-Arg, and L-Gln present in Abound^®^ are speculated to promote wound healing through the synthesis of proteins, such as hydroxyproline, a collagen precursor necessary for epithelialization, and granulation tissue proliferation, fibrosis, and inhibition of proteolysis [31]. The key enzymes that promote protein synthesis include Gln, vitamins, Arg, branched chain amino acid (BCAA), zinc, phosphorus, and omega-3 fatty acids, which are known to be useful for various pathologies. In a meta-analysis, Zheng et al. have reported the usefulness of Gln administration in terms of cumulative nitrogen balance, infectious complications, and postoperative hospital stay in abdominal surgery [32]. Arg reportedly promotes protein synthesis and nitric oxide production. Moreover, Arg is thought to be a substrate for nitric oxide, which has a vasodilatory effect, and to play a role in maintaining microcirculation at the site of gastrointestinal anastomosis [5,33]. BCAAs are composed of three amino acids (valine, leucine, and isoleucine) with branched side chains [34], and they are known to have the strongest protein anabolic effect among the essential amino acids [35,36]. In liver cirrhosis, protein assimilation does not function properly, especially in a state of leucine deficiency, but BCAA administration is expected to suppress ammonia synthesis and improve the efficiency of nitrogen utilization in the body [35,36,37]. In the treatment of liver cirrhosis patients, the usefulness of BCAA administration and exercise therapy has been reported for sarcopenia [24,38,39]. The importance of nutritional therapy and rehabilitation in the perioperative period of esophageal cancer surgery has also been demonstrated [25,27]. Zinc consumption is increased in highly invasive pathological conditions, and it promotes muscle breakdown, causes anemia, and affects the hormones [40,41]. ω3 fatty acids are useful in maintaining the nutritional status of patients who had undergone gastrointestinal cancer surgery through their anti-inflammatory effects [42]. Calcium HMB is a metabolic product of leucine and has the effect of promoting protein synthesis via the mammalian target of rapamycin pathway and inhibiting body protein breakdown via the inhibition of the ubiquitin-proteasome system. The administration of HMB synergistically enhances the metabolic improvement in leucine synthesis in skeletal muscles [35,37,43,44]. Additionally, HMB promotes wound healing together with Gln, which is an energy substrate for cells involved in wound healing, including macrophages, lymphocytes, and fibroblasts [6,31,45]. Theoretically, our study results suggest that sufficient Gln, Arg, and HMB administration can counteract the inhibitory effect of postoperative protein synthesis.

The second cause of sarcopenia progression is the relative lack of nutrients due to increased inflammation after surgery. Esophageal cancer surgery causes widespread tissue damage, and for some period of time after surgery, exudates are drained out of the body, causing a large amount of protein leakage. The amount of protein lost cannot be replenished by the nutrients administered alone; thus, the body dissolves its own skeletal muscles to mobilize the protein components into the plasma, which is expected to cause considerable atrophy of the skeletal muscles during this period [3,4,21]. It is expected that administering Gln—a material for skeletal muscles—before surgery, prior to protein wasting during this period, may suppress the development of muscle atrophy in the early postoperative phase. Oral Gln administration also enhances immunity, prevents bacterial translocation, and improves the survival rates in patients with severe infections [5,45,46]. The nutritional management guidelines recommend a daily Gln dose of 0.3–0.5 g/kg/day, and the Gln/Arg/HMB combination used in this study contains 14 g of Gln and 14 g of Arg per two packets, which is thought to meet this recommended dose [6,45,47,48]. Contrarily, Orlila, et al. reported that in the perioperative nutritional therapy for malnourished patients, the preoperative intravenous administration is more effective in reducing the incidence of postoperative complications and length of stay in the intensive care unit as compared to enteral administration [49]. Ayala et al. also reported that intravenous Gln administration relieves surgery-related immunosuppression by suppressing interferon (IFN)-γ and interleukin (IL)-4 productions early in response to the overexpression of IFN-γ and IL-4 from T lymphocytes in response to surgical stress [50,51]. Unfortunately, intravenous Gln preparations have not been approved in Japan; thus, we could not use it. However, enteral administration alone can provide some benefits from systemic distribution via intestinal absorption.

The third reason is that postoperative wound pain, fatigue, and general exhaustion reduce an individual’s ability to perform ADL, which impedes early postoperative rehabilitation. Perioperative nutritional management tailored to the patient’s condition has been reported to promote wound healing and maintain overall health [6]. According to the European Society of Clinical Nutrition and Metabolism (ESPEN) and American Society for Parenteral and Enteral Nutrition (ASPEN) guidelines, early initiation of enteral nutrition after gastrointestinal cancer surgery is strongly recommended and has become widespread as part of ERAS, but a definitively useful nutritional management method has yet to be established [52,53]. Based on the ERAS concept, rehabilitation programs aimed at maintaining ADL throughout the entire perioperative period have become widely practiced in clinical practice [25,26,54,55]. The contents of ERAS include omission of preoperative intestinal preparation, shortening of the fasting period, oral care, routine postoperative analgesia, early mobilization, and initiation of early postoperative enteral nutrition [25,27,54,55]. Atrophy of the intestinal mucosa is closely related to Gln deficiency. Gln needs to be sufficiently distributed and utilized by the intestinal epithelium to regenerate the atrophied intestinal mucosa. By preventing mucosal atrophy, bacterial translocation can be suppressed to maintain an individual’s immune function [46,56]. Administering sufficient amounts of Gln, together with early enteral nutrition, is an efficient approach that is in line with the concept of ERAS. There have also been reports on the synergistic effect of early rehabilitation and Gln in increasing the skeletal muscle mass, and this is considered a rehabilitation nutritional approach that can be applied to prevent disuse syndrome in various surgical patients [36,39,55]. In our department, we also practice ERAS in the perioperative management of gastrointestinal cancer, including the appropriate use of analgesics, dental intervention, and early enteral nutrition, and the good results of this study are thought to be the overall result of these efforts [57]. However, from the perspective of preventing postoperative chylothorax, a previously utilized fat-containing enteral nutrition was changed to a fat-free nutritional supplement in the later cases, which may have had an impact. If carbohydrates are administered as enteral nutrition after surgery, there is a concern that they may cause dumping syndrome due to their rapid administration into the intestinal tract; however, our nutritional supplement preparation contains almost no carbohydrates and is mainly composed of amino acids, so it is unlikely to cause rapid blood sugar fluctuations and is theoretically unlikely to cause dumping syndrome. Motoori et al. have reported the usefulness of immunonutrition and synbiotics in perioperative and preoperative chemotherapy for esophageal cancer; they also demonstrated that they contribute to the normalization of the intestinal flora and reduction of chemotherapy-induced adverse events and postoperative complications [58]. Additionally, Mayanagi et al. have shown that skeletal muscle atrophy caused by preoperative chemotherapy was reduced after surgery [59]. There have been no reports to date on the usefulness of Gln, Arg, or HMB in preoperative chemotherapy for esophageal cancer, which require further investigation in order to improve the treatment outcomes of patients with esophageal cancer.

In the present study, sarcopenia was defined as having a low PMI, but sarcopenia should be defined using multiple factors, including walking speed, grip strength, upper arm circumference, and body composition analysis through a bioimpedance analysis [60]. However, the present investigation was a retrospective study, and various parameters related to the definition of sarcopenia were missing and could not be used. Our study subjects were patients who had undergone MIE, and measuring PMA using CT images before and after surgery was possible; thus, for convenience, we used CT image to define sarcopenia. In the present study, the cutoff value for diagnosing sarcopenia based on the iliopsoas muscle area was adopted as an index appropriate for the actual situation of Japanese people, as reported by Hamaguchi et al. [11].

Our study results showed that the incidence of postoperative pleural effusion and expectoration disorders was significantly lower in the Gln/Arg/HMB group than in the control group. This may be due to the prevention of the development of sarcopenia in the respiratory muscles and maintenance of colloid osmotic pressure as a result of promoting protein synthesis, which is similar to the results of a previous study [61]. Additionally, from our personal experience, the administration of a Gln/Arg/HMB product to patients with anastomotic leakage after esophageal cancer surgery shortened the recovery period from anastomotic leakage (13.5 ± 14.3 vs. 35.0 ± 17.4 days, *p* = 0.043) and maintained serum albumin levels on 14th postoperative day (−1.0 ± 0.32 vs. −1.4 ± 0.49 g/dL, *p* = 0.047), suggesting its usefulness in wound healing and the maintenance of nutritional status.

Among the postoperative nutritional indicators, body weight and PMA were significantly higher in the Gln/Arg/HMB group at 1 week and 1 month postoperatively. Only a few studies have reported significant changes in the parameters after the perioperative nutritional intervention [62,63]; thus, the results of this study are considered to be very important findings. However, further study is still needed to determine the detailed impact of this intervention on the nutritional parameters.

The occurrence of complications after esophageal cancer surgery has a great impact on a patient’s long-term prognosis [64,65,66,67]. It has been suggested that various inflammatory cytokines and chemokines, such as IL-6 and IL-8, not only promote cancer cell proliferation and activation but may also synergistically promote cancer progression through the suppression of natural killer cell function. The transition of perioperative inflammatory cytokines, which is enhanced in cases with postoperative complications, may be deeply involved in the long-term prognosis [62,63,68]. This highlights the importance of the need for preventing complications.

A limitation of this study is that the historical backgrounds of the two groups differed. It is possible that the postoperative outcome is influenced not only by nutritional therapy but also by the progress of postoperative management. There may have been some differences between the two groups in the perioperative management of patients using ERAS, the response to postoperative complications, and the fundamental nutritional supplements used, as well as differences in the quality of surgery. Additionally, although no apparent adverse events occurred due to the administration of Gln/Arg/HMB product in the cases examined here, diarrhea or abdominal pain may occur depending on the administration dosage and speed, and it is unclear whether the Gln/Arg/HMB product can be administered to all cases. Furthermore, in cases of renal or liver dysfunction, excessive administration of amino acids may lead to the deterioration of renal or liver function; thus, dosage adjustments will be necessary depending on the case. Additionally, it is expected that the administration of Gln/Arg/HMB product can reduce the occurrence of complications, which may ultimately extend the patients’ prognosis, but we were unable to follow up on the long-term outcomes in this study. This is an issue that needs to be investigated in the future.

## 5. Conclusions

Even with the introduction of MIE, radical surgery for esophageal cancer remains a great burden. Postoperative complications significantly impact esophageal cancer patients, resulting in malnutrition, sarcopenia, reduced QOL, and poor prognosis. Therefore, perioperative management, including nutritional therapy using disease-specific Gln/Arg/HMB products, may prevent the onset of complications and minimize their impact, leading to improved short-term postoperative outcomes, skeletal muscle mass, and QOL. To improve survival, it is crucial to provide appropriate perioperative nutritional support specifically tailored to the disease. Future studies should focus on developing evidence for optimal nutritional interventions, and exhibiting the impact of amino acid-enriched nutritional support.

## Figures and Tables

**Table 1 nutrients-16-02126-t001:** Clinico-oncological characteristics and physical parameter values of the patients in the control and the Gln/Arg/HMB groups.

		Control Group (n = 79)	Gln/Arg/HMB Group (n = 114)	*p* Value
Age (years)	Mean ± SD	64.5 ± 8.3	65.6 ± 7.8	0.354 ^a^
Sex	Male/Female	62/17	92/22	0.706 ^b^
PS (ECOG)	0/1≤	77/2	103/11	0.078 ^c^
Tumor location	Upper/Middle/Lower	11/44/24	15/55/44	0.492 ^b^
Histology	SCC/AC/Others	76/3/0	105/6/3	0.667 ^b^
fT (UICC 8th)	is/1/2/3/4	0/31/7/27/14	1/47/13/30/23	0.708 ^b^
fN (UICC 8th)	0/1/2/3	40/22/12/5	40/41/21/12	0.181 ^b^
cM (UICC 8th)	0/1	68/11	98/16	0.983 ^b^
fStage (UICC 8th)	0/I/II/III/IVA/IVB	11/12/21/22/12/1	10/19/31/29/22/3	0.824 ^b^
Preoperative chemotherapy	Present/Absent	38/41	71/43	0.051 ^b^
Height (cm)	Mean ± SD	162.8 ± 7.8	164.1 ± 7.4	0.243 ^a^
Body weight (kg)	Mean ± SD	58.6 ± 9.9	58.9 ± 9.5	0.842 ^a^
BMI	Mean ± SD	21.9 ± 2.8	21.5 ± 2.8	0.297 ^a^
BSA (m^2^)	Mean ± SD	1.62 ± 0.18	1.62 ± 0.15	0.965 ^a^
PMA (cm^2^)	Mean ± SD	1527.2 ± 479.0	1519.6 ± 440.2	0.910 ^a^
PMI (cm^2^/m^2^)	Mean ± SD	5.69 ± 1.56	5.61 ± 1.53	0.716 ^a^
Preoperative sarcopenia	Present/Absent	42/37	66/48	0.515 ^b^
The count of WBC (/mm^3^)	Mean ± SD	5115 ± 1317	5416 ± 1650	0.178 ^a^
The count of neutrophil (/mm^3^)	Mean ± SD	2875 ± 1009	3225 ± 1354	0.053 ^a^
The count of lymphocyte (/mm^3^)	Mean ± SD	1702 ± 522	1604 ± 525	0.203 ^a^
CRP	(mg/dL)	0.28 ± 0.68	0.26 ± 0.54	0.846 ^a^
Total protein	(g/dL)	6.56 ± 0.52	6.59 ± 0.54	0.746 ^a^
Serum albumin	(g/dL)	3.95 ± 0.32	3.92 ± 0.37	0.549 ^a^
Total cholesterol	(mg/dL)	180.3 ± 48.0	172.8 ± 46.1	0.288 ^a^
Triglycerides	(mg/dL)	118.9 ± 72.0	111.7 ± 58.2	0.462 ^a^

Gln, Glutamine; Arg, Arginine; HMB, β-hydroxy-β-methylbutyrate; SD, standard deviation; PS, physical status; SCC, squamous cell carcinoma; AC, adenocarcinoma; UICC 8th, the Union for International Cancer Control TNM staging, version 8; BMI, body mass index; BSA, body surface area; PMA, Area of psoas muscle at the L3 level; PMI, psoas muscle index; WBC, white blood cell; CRP, C-reactive protein; ^a^ Student’s *t*-test; ^b^ χ^2^ test, ^c^ Fisher’s test.

**Table 2 nutrients-16-02126-t002:** Intraoperative factors of the control and Gln/Arg/HMB group.

		Control Group (n = 79)	Gln/Arg/HMB Group (n = 114)	*p* Value
Total operation time (min)	Mean ± SD	633 ± 117	640 ± 99	0.647 ^a^
Total blood loss (g)	Mean ± SD	672 ± 567	466 ± 318	* 0.002 ^a^
Intrathoracic approach	Thoracoscopic/Robotic surgery	79/0	102/12	* 0.002 ^b^
Abdominal approach	Open/HALS	16/63	38/76	* 0.047 ^c^
Reconstructive organ	Stomach/Jejunum	74/5	106/8	0.851 ^c^
Reconstruction route	Mediastinal/Retrosternal	75/4	102/12	0.197 ^b^
Field of lymphnode dissection	D2/D3	25/54	45/69	0.266 ^c^
Resectability	R0/R1≤	73/6	103/11	0.621 ^c^

Gln, Glutamine; Arg, Arginine; HMB, β-hydroxy-β-methylbutyrate; SD, standard deviation; HALS, hand-assisted laparoscopic surgery; * *p* < 0.05. ^a^ Student’s *t*-test; ^b^ Fisher’s exact test; ^c^ χ^2^ test.

**Table 3 nutrients-16-02126-t003:** Postoperative complications after MIE (Clavien Dindo grade ≥ III).

	Control Group (n = 79)	Gln/Arg/HMB Group (n = 114)	*p* Value
All complications	49 (62.0%)	44 (38.6%)	* 0.001 ^a^
Recurrent nerve palsy	3 (3.8%)	6 (5.3%)	0.740 ^b^
Atelectasis, sputum expectoration disorder	35 (44.3%)	32 (28.1%)	* 0.020 ^a^
Pneumonia	4 (5.1%)	10 (8.8%)	0.406 ^b^
ARDS, respiratory failure	6 (7.6%)	10 (8.8%)	0.771 ^a^
Pleural effusion	22 (27.8%)	15 (13.2%)	* 0.011 ^a^
Chylothorax	1 (1.3%)	2 (1.8%)	>0.999 ^b^
Anastomotic leakage	6 (7.6%)	9 (7.9%)	0.939 ^a^
SSI	14 (17.7%)	11 (9.6%)	0.101 ^a^
ACS, heart failure	3 (3.8%)	0	0.067 ^b^
Anastomotic stenosis	20 (25.3%)	27 (23.7%)	0.795 ^a^
Re-operation	2 (2.5%)	5 (4.4%)	0.702 ^b^
Postoperative hospital stay (days)	46.3 ± 29.6	49.5 ± 57.3	0.659 ^c^

Numbers given as n (%). MIE, minimally invasive esophagectomy; Gln, Glutamine; Arg, Arginine; HMB, β-hydroxy-β-methylbutyrate; ARDS, acute respiratory distressed syndrome; SSI, surgical site infection; ACS, acute coronary syndrome; * *p* < 0.05. ^a^ χ^2^ test; ^b^ Fisher’s exact test; ^c^ Student’s *t*-test.

**Table 4 nutrients-16-02126-t004:** The changes in body weight and PMA after MIE.

	Control Group (n = 79)	Gln/Arg/HMB Group (n = 114)	*p* Value
Body weight (kg)			
Before operation	58.6 ± 9.9	58.9 ± 9.5	0.842 ^a^
After 1 month	54.8 ± 9.4	56.0 ± 8.4	0.363 ^a^
After 1 year	50.9 ± 8.8	52.3 ± 8.0	0.237 ^a^
The change rate of body weight (%)			
After 1 month	93.6 ± 3.1	95.3 ± 3.5	* <0.001 ^a^
After 1 year	87.1 ± 7.9	89.3 ± 6.9	* 0.047 ^a^
PMA (mm^3^)			
Before operation	1527 ± 479	1520 ± 440	0.910 ^a^
After 1 month	1420 ± 440	1508 ± 413	0.157 ^a^
After 1 year	1402 ± 448	1441 ± 412	0.536 ^a^
The change rate of PMA (%)			
After 1 month	93.5 ± 7.5	99.9 ± 6.5	* <0.001 ^a^
After 1 year	92.0 ± 7.9	95.4 ± 8.7	* 0.006 ^a^

PMA, psoas muscle mass; MIE, minimally invasive esophagectomy; Gln, Glutamine; Arg, Arginine; HMB, β-hydroxy-β-methylbutyrate; * *p* < 0.05. ^a^ Student’s *t*-test.

**Table 5 nutrients-16-02126-t005:** Univariate and multivariate analyses of clinicopathological variables associated with overall survival.

Variables		Univariate Analysis			Multivariate Analysis		
		HR	95% CI	*p*-Value	HR	95% CI	*p*-Value
Sex	Female vs. Male	1.870	0.963–3.630	0.064			
Preoperative chemotherapy	Present vs. Absent	1.023	0.656–1.594	0.921			
PS	0 vs. 1≤	1.401	0.607–2.882	0.597			
Abdominal procedure	HALS vs. Open	1.269	0.764–2.105	0.358			
fT	0–2 vs. 3–4	1.953	1.247–3.067	* 0.003			
fN	0 vs. 1≤	2.273	1.397–3.690	* 0.001	2.364	1.449–3.861	* 0.001
cM	0 vs. 1	1.206	0.666–2.188	0.536			
fStage	0–2 vs. 3–4	2.778	1.751–4.425	* <0.001			
Resectability	R0 vs. R1≤	5.917	3.311–10.526	* <0.001	6.410	3.546–10.204	* <0.001
The count of lymphocyte	1200/mm^3^≤ vs. <1200/mm^3^	1.292	0.713–2.341	0.399			
CRP	<0.5 mg/dL vs. 0.5 mg/dL≤	1.961	1.117–3.436	* 0.019			
Serum albumin	3.5 mg/dL≤ vs. <3.5 mg/dL	1.042	0.453–2.397	0.922			
Preoperative sarcopenia	Absent vs. Present	1.014	0.651–1.580	0.952			
Postoperative complications	Absent vs. Present	1.290	0.829–2008	0.259	1.416	0.908–2.212	0.125
Use of Gln/Arg/HMB product	Absent vs. Present	0.929	0.595–1.453	0.748			

CI, confidential interval; PS, physical status; HALS, hand assisted laparoscopic surgery; CRP, C-reactive protein; Gln, Glutamine; Arg, Arginine; HMB, β-hydroxy-β-methylbutyrate; * *p* < 0.05. Variables were adjusted for in the multivariable logistic regression model.

## Data Availability

The datasets used and analyzed during the current study are available from the corresponding author on reasonable request.

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
