# Peer review of "Usefulness of Perioperative Nutritional Therapy with the Glutamine/Arginine/Calcium β-Hydroxy-β-Methylbutyrate Product in Esophageal Cancer Surgery: A Single-Center Retrospective Study"

_nutrients, 2024, doi:10.3390/nu16132126_

Round 1

Reviewer 1 Report

Comments and Suggestions for Authors

This study aimed to investigate whether the postoperative outcomes, including complication rate, nutritional status, and skeletal muscle mass, can be improved with amino acid-rich perioperative nutritional therapy using glutamine (Gln)/arginine (Arg)/calcium β-hydroxy-β-methylbutyrate (HMB) products as pharmaconutrition in the perioperative management of patients who had undergone esophageal cancer surgery. Based on their findings, it concluded that perioperative amino acid-enriched nutritional therapy may improve the short-term postoperative outcomes, nutritional status, and skeletal muscle mass of esophageal cancer surgical patients. 

Major comment

1. Please describe how to identify the control group.

2. I agree with the authors about the potential baseline differences between the study and group, such as the introduce of ERAS or other intervention during the study period. The authors may make it more clear in the limitation section.

Author Response

Dear Editor and reviewers:

Thank you very much for taking the time to review this manuscript. Please find attached a revised version of our manuscript “Usefulness of Perioperative Nutritional Therapy with the Glutamine/Arginine/Calcium β-Hydroxy-β-Methylbutyrate Product in Esophageal Cancer Surgery: A Single-Center Retrospective Study”, which we would like to resubmit for publication as an article in Nutrients.

In response to the suggestions of editor and reviewers, we have corrected the revised manuscript as possible as we can respond. We ensured the revised manuscript with all changes clearly highlighted.
We hope that the revisions in the manuscript and our accompanying responses are sufficient to make our manuscript suitable for publication in Nutrients.

We look forward to hearing from you at your convenience.

Reviewer 2 Report

Comments and Suggestions for Authors

Dear Authors,

thank you for reminding us the Abound, an all-time-classic supplement!

I have one question only - material is somewhere confused or perhaps there is mistake in the dates you are referred to

You have an historic control group [Sep 2004 to May 2012]

This group received enteral MEIN [having fat] from Jan 2003??? to March 2013

The Abound group started May 2012

that means there are some participants who received MEIN and Abound [from May 2012 up to April 2013] and the rest participants received Elental [no-fat] and Abound.

I hope your are a little confused with the dates - otherwise you should  exclude the Abound-patients who have received fat diet OR to split the group [fat and no-fat subgroups plus Abound] OR to explain [difficult] if there were any differences in results, and include this in limitations.

Author Response

(The authors gave the same response as above.)
